# Leveraging Double Descent for Scientific Data Analysis: Face-Based Social Behavior as a Case Study

## Abstract

Scientific data analysis often involves making use of a large number of correlated predictor variables to predict multiple response variables. Understanding how the predictor and response variables relate to one another, especially in the presence of relatively scarce data, is a common and challenging problem. Here, we leverage the recently popular concept of "double descent" to develop a particular treatment of the problem, including a set of key theoretical results. We also apply the proposed method to a novel experimental dataset consisting of human ratings of social traits and social decision making tendencies based on the facial features of strangers, and resolve a scientific debate regarding the existence of a "beauty premium" or "attractiveness halo," which refers to a (presumed) advantage attractive people enjoy in social situations. We demonstrate that more attractive faces indeed enjoy a social advantage, but this is indirectly due to the facial features that contribute to both perceived attractiveness and trustworthiness, and that the component of attractiveness perception due to facial features (unrelated to trustworthiness) actually elicit a "beauty penalty.". Conversely, the facial features that contribute to trustworthiness and not to attractiveness still contribute positively to pro-social trait perception and decision making. Thus, what was previously thought to be an attractiveness halo/beauty premium is actually a trustworthiness halo/premium plus a "beauty penalty." Moreover, we see that the facial features that contribute to the trustworthiness halo primarily have to do with how smiley a face is, while the facial features that contribute to attractiveness but actually acts as a beauty penalty is related to anti-correlated with age. In other words, youthfulness and smiley-ness both contribute to attractiveness, but only smiley-ness positively contributes to pro-social perception and decision making, while youthfulness actually negatively contribute to them. A further interesting wrinkle is that youthfulness as a whole does not negatively contribute to social traits/decision-making, only the component of youthfulness contributing to attractiveness does.

## 1 Introduction

Scientific data analysis often involves building a linear regression model between a large number of predictor variables and multiple response variables. Understanding how the predictor and response variables relate to one another, especially in the presence of relatively scarce data, is an important but challenging problem. For example, a geneticist might have a genomic dataset with many genetic features as predictor variables and disease prevalence data as response variables: the geneticist may want to know how the different types of disease are related to each other through their genetic underpinnings. Another example is that a social psychologist might have a set of face images (with many facial features) that have been rated by a relatively small set of subjects for perceived social traits and social decision making tendencies, and wants to discover how the different social traits and decision making tendencies relate to each other through the underlying facial features.

A common problem encountered in these types of problems is that the large number of features relative to the number of data points typically entails some kind of dimensionality reduction and feature selection, and this process needs to be differently parameterized in order to optimize for each response variable, making direct comparison of the features underlying different response variables

challenging. In the worst case, there may not be *any* subset of features that can predict all response variables better than chance level. Here, we leverage the "double descent" phenomenon to develop and present a novel analysis framework that obviates such issues by relying on a universal, overly parameterized feature representation. As a case study, we apply the framework to better understand the underlying facial features that contribute separately and conjointly to human trait perception and social decision making.

Humans readily infer social traits, such as attractiveness and trustworthiness, from as little as a 100 ms exposure to a stranger's face (Willis & Todorov, 2006). Though the veracity of such judgments is still an area of active research (Valla et al., 2011; Todorov et al., 2015), such trait evaluations have been found to predict important social outcomes, ranging from electoral success (Todorov et al., 2005; Ballew & Todorov, 2007; Little et al., 2007) to prison sentencing decisions (Blair et al., 2004; Eberhardt et al., 2006).

In particular, psychologists have observed an "attractiveness halo", whereby humans tend to ascribe more positive attributes to more attractive individuals (Eagly et al., 1991; Langlois et al., 2000), and economists have observed a related phenomenon, the "beauty premium", whereby more attractive individuals out-earn less attractive individuals in economics games (Mobius & Rosenblat, 2006). However, these claims are not without controversy (Andreoni & Petrie, 2008; Willis & Todorov, 2006), as more attractive people can also incur a "beauty penalty" in certain situation. Moreover, a robust correlation between attractiveness and trustworthiness (Willis & Todorov, 2006; Oosterhof & Todorov, 2008; Xu et al., 2012; Ryali et al., 2020) has also been reported, making it unclear how much of the attractiveness halo effect might be indirectly due to perceived trustworthiness.

To tease apart the contributions of trustworthiness and attractiveness to social perception and decision-making, we perform linear regression of different responses variables, consisting of subjects' ratings of social perception and social decision-making tendencies, against features of the Active Appearance Model (AAM), a well-established computer vision model (Cootes et al., 2001), whose features have been found to be linearly encoded by macaque face-processing neurons (Chang & Tsao, 2017). A similar regression framework has been adopted by previous work modeling human face trait perception (Oosterhof & Todorov, 2008; Said & Todorov, 2011; Song et al., 2017; Guan et al., 2018; Ryali et al., 2020), using features either from AAM or deep neural networks. Because the number of features is typically quite large, usually larger than the number of rated faces, previous approaches have all used some combination of dimensionality reduction and feature selection. This approach gives rise to a dilemma when one wants to compare the facial features contributing to different types of social perceptions (response variables), since the number of features that optimizes prediction accuracy for each task can be quite different (see Figure 1). Either one optimizes this quantity separately for each task, thus not having a common set of features to compare across; or one can fix a particular set of features for all tasks, but then having suboptimal prediction accuracy (in the worst case, perhaps worse than chance level performance).

To overcome this challenge, we appeal to 'the "double descent" (Belkin et al., 2019; 2020) trick, the use of a highly overparameterized representation (more features than data points) to achieve good performance. In particular, if we use the original AAM feature representation, while foregoing any kind of dimensionality reduction or feature selection, then we have a universal representation that may also have great performance on all tasks, even novel tasks not seen before, or responses corresponding to predictor variable settings totally different than previously seen. While overparameterized linear regression has chiefly been used as an analytically tractable case study (Belkin et al., 2019; Xu & Hsu, 2019; Belkin et al., 2020) to gain insight into the theoretical basis and properties of "double descent", we use it as a practical setting for scientific data analysis. Notably, while previous papers on overparameterized regression defined statistical assumptions and constraints in the generative sense, we work for pragmatic reasons purely with sample statistics (e.g. whether two features are "truly" decorrelated (Xu & Hsu, 2019)), we work directly with sample statistics (e.g. whether two feature vectors across a set of data points have a correlation coefficient of 0). For this reason, our theoretical results are distinct from and novel with respect to those prior results. Finally, it is noteworthy that the human visual pathway also exhibits feature expansion rather than feature reduction, from the sensory periphery to higher cortical areas (Wandell, 1995) – this raises the intriguing possibility that the brain has also discovered an overparameterized representation as a universal representation for learning to perform well on many tasks, including novel ones not previously encountered.

In Section 2, we use over-parameterized linear regression to develop a framework that generalizes well across both tasks and data space. We provide theoretical conditions under which the complete over-parameterized representation is 1) guaranteed to yield linear estimators that perform better than chance, and 2) are optimal among the class of hard regularizers. We also provide exact error expressions for these estimators, as well as an exact measure of how far the estimators are from the optimal hard regularizers when the over-parameterized estimators are suboptimal.

In Section 3, we verify the practical usefulness of our theoretical framework by comparing the prediction accuracy of over-parameterized regression against task-specific classical (under-parameterized) regression that optimizes feature selection for each task, on original data collected from a face-based social perception and decision-making study.

In Section 4, we apply our mathematical and computational framework to show that the halo effect appears to arise from trustworthiness rather than attractiveness *per se*, and that attractiveness unrelated to trustworthiness actually induces a beauty *penalty*, while trustworthiness unrelated to attractiveness induces a premium, thus reconciling conflicting results in the literature regarding the existence of an attractiveness halo. Finally, we present a novel finding that the component of attractiveness related to pro-social perception and judgment is related to how smiley a face appears, while the component of attractiveness *unrelated* to attractiveness is related to the youthfulness of facial appearance.

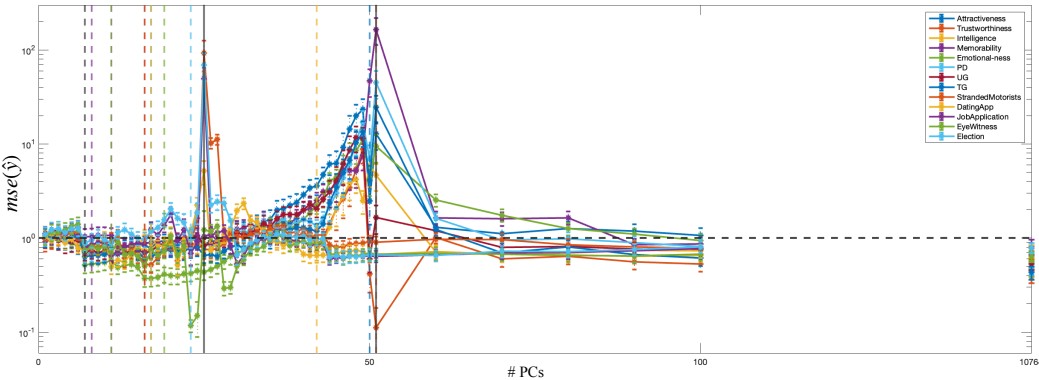

Figure 1: Loss (prediction MSE), as a function of number of features for face-based social-perception tasks, using cross-validation as specified in section 3. The vertical dashed lines indicate the minimum error in the under-parameterized regime for the different tasks (repsonse variables), illustrating the difficulty of finding a common number of features to use for all tasks in the under-parameterized regime. The fully over-parameterized regression results in better-than-chance MSE for all tasks. Horizontal dashed line: variance of collected responses, normalized to 1 in all tasks for ease of visual comparison; error bars: standard error of the mean.

## 2 MATHEMATICAL FRAMEWORK

We consider a linear regression problem where each response $y$ is a linear function of $n$ real-valued variables $\mathbf{x} \in \mathbb{R}^n$, parameterized by a vector $\beta \in \mathbb{R}^n$, in addition to some noise ($\epsilon$). More formally for $m$ datapoints (with $n \geq m$), we assume:

$$\mathbf{y} = \mathbf{X}\beta + \epsilon, \qquad \epsilon \sim \mathcal{N}(0, \sigma_\epsilon^2). \tag{1}$$

with both $\beta$ and $\epsilon$ zero-mean and i.i.d.. We further assume, without loss of generality, that both the design matrix $\mathbf{X} \in \mathbb{R}^{m \times n}$ and the vector of responses $\mathbf{y} \in \mathbb{R}^m$ are centered, and that $\mathbf{X}$ is full rank, with rank denoted by $r$. Note that for an over-parameterized, centered full rank matrix, $r = m - 1$.

We use the pseudoinverse to obtain the $n$-dimensional minimum L2-norm estimator $\hat{\beta}$ of $\beta$,

$$\hat{\beta} = \mathbf{X}^\dagger \mathbf{y}, \tag{2}$$

and the mean squared error (MSE) to evaluate the estimator $\hat{\beta}$:

$$\text{mse}(\hat{\beta}) = \text{tr}\,\mathbb{E}[(\beta - \hat{\beta})(\beta - \hat{\beta})^T], \tag{3}$$

where $\text{tr}(\cdot)$ denotes the matrix trace and $\mathbb{E}[\cdot]$ the expected value. We also use $||\cdot||$ to denote the L2-norm.

## 2.1 THEORETICAL CONDITIONS FOR A GOOD ESTIMATOR $\hat{\beta}$

**Estimator Condition 1.** If the noise-variance in $\mathbf{y}$ $(\sigma_\epsilon^2)$ is less than or equal to half the signal-variance in $\mathbf{y}$ $(\sigma_y^2)$, then $\hat{\beta}$ is an above chance estimator, i.e.

$$\sigma_\epsilon^2 \leq \frac{\sigma_y^2}{2} \quad \Longleftrightarrow \quad \text{mse}(\hat{\beta}) \leq \sigma_\beta^2, \tag{4}$$

where $\sigma_\beta^2$ is the variance of the parameter vector $\beta$.

*Proof Sketch.* This follows from the definition of $\text{mse}(\hat{\beta})$ and $\sigma_\beta^2$, the cyclical properties of the trace and linearity of expectation, as well as the i.i.d. assumptions on $\beta$ and $\epsilon$. See the appendix for an explicit derivation.

**Estimator Condition 2.** If the smallest singular value $(s_r)$ of the design matrix $\mathbf{X}$ satisfies,

$$s_r^2 \geq \frac{\sigma_\epsilon^2}{||\beta||^2/r}, \tag{5}$$

then $\hat{\beta}$ is the minimum MSE (MMSE) estimator among the class of hard regularizers subject to linear constraints.

*Proof Sketch.* Park (1981) proved the above for the prediciton MSE of a PCR estimator in the under-parameterized regime. An extension to the over-parameterized regime follows as 1) the MSE is invariant under orthogonal transformations, and 2) any over-parameterized estimator has an "equivalent" under-parameterized estimator (equivalent in the sense that the estimators yield the same MSE). See the appendix for a detailed proof.

## 2.2 EXACT ERROR EXPRESSIONS

**Error Expression 1.** The MSE of the over-parameterized estimator $\hat{\beta}$ is given by

$$\text{mse}(\hat{\beta}) = \sigma_\epsilon^2 \sum_{i=1}^{r} \frac{1}{s_i^2}, \tag{6}$$

where $s_1, ..., s_r$ are the singular values of the design matrix $\mathbf{X}$.

*Proof Sketch.* Once again, this follows from the definition of $\text{mse}(\hat{\beta})$, the cyclical properties of the trace and linearity of expectation, as well as the i.i.d. assumptions on $\beta$ and $\epsilon$. See the appendix for an explicit derivation.

**Error Expression 2.** Suppose the MMSE estimator $\hat{\theta}^*$ has $p$ components. Then the difference in MSEs between the MMSE estimator and the fully over-parameterized estimator is given by,

$$\text{mse}(\hat{\beta}) - \text{mse}(\hat{\theta}^*) = \sigma_\epsilon^2 \sum_{i=p+1}^{r} \frac{1}{s_i^2} - \frac{||\beta||^2}{r}(r - p). \tag{7}$$

*Proof Sketch.* This follows from extending Park (1981) to the over-parameterized regime, in addition to the definition of MSE, the cyclical properties of the trace and linearity of expectation, as well as the i.i.d. assumptions on $\beta$ and $\epsilon$. See the appendix for a detailed proof.

## 3  EXPERIMENTAL VALIDATION & COMPUTATIONAL FRAMEWORK

As the theoretical conditions and error expressions established in the previous section depend on variables that are unknown in real data (such as noise and signal variance, the norm of the true parameter vector etc.), and as real data may violate the theoretical assumptions, we validate how well the over-parameterized representation generalizes in practice using data collected in a face-based social decision-making study (Figure 2).

### 3.1  SOCIAL DECISION-MAKING EXPERIMENT

613 undergraduate students at the University of California, San Diego participated in a 3 block hour long study in which they were asked to rate social traits (block A), make decisions in social scenarios (block B), and play economic games (block C) with novel face images (Figure 2). All blocks were counterbalanced across subjects.

**Inclusion/exclusion criteria.** Participants who had a response entropy and/or a CC between their response and the average response below two standard deviations of the mean were excluded, resulting in standardized responses from 485 subjects being included in the analysis.

**Face stimuli.** 72 white female faces with direct gaze and natural expressions were sampled from the 10K US Adult Faces Database (Bainbridge et al., 2013). A sub-sample of 52 faces was then used in blocks A and C, while 36 face pairings were used in block B. The 52 face images used in all blocks were included in the analysis.

**A. Social Decision Making Tasks**

| Traits | • Attractive • Dominant • Emotional • Intelligent • Memorability • Trustworthy |
| Social Scenarios | • Dating App • Job Interview • Eyewitness • Election • Standard Motorist |
| Economic Games | • Trust Game • Ultimatum Game • Prisoner's Dilemma |

**B. Social Scenarios**

1. **Dating App.** Suppose two people on a dating app sent you a greeting message. Which of them would you (or your friend of the appropriate sexual orientation) be more willing to respond to?

2. **Job Interview.** Suppose you represent a company at a job fair, and two individuals approached you to discuss job openings. Which of them would you more willing to talk to?

3. **Eye Witness.** Suppose two people are eye witnesses of a gas station robbery and gave contrary accounts. Which of them would you be more willing to believe?

4. **Election.** Suppose two people are candidates for a state-wide election. Which of them are you more likely to vote for?

5. **Stranded Motorist.** Suppose two drivers are standing next to broken-down cars on the side of the highway. Which of them would you be more willing to help?

Figure 2: Overview of the face-based social decision-making experiment. **(A)** The trait rating tasks (block A), social scenario tasks (block B) and economic games (block C) with sample screenshots from the experiment display. For each task, participants respond on a scale of 1-9. For the social scenario tasks, 1/9 indicates maximal preference for the face on the left/right, while 5 indicates equally preferable. In Prisoner's Dilemma (PD; Kremp et al., 1982) participants are asked how likely they are to cooperate (rather than defect). In the Ultimatum Game (UG; Solnick and Schweitzer, 1999) and Trust Game (TG; Wilson and Eckel, 2006) participants are asked how much money (in $) they would invest (TG) or propose (UG). **(B)** Questions displayed in the five social scenario tasks.

## 3.2 COMPUTATIONAL FRAMEWORK

**Feature Representation.** We train a three-color-channel AAM on the Chicago Face Database (Ma et al., 2015) plus the 10K US Adult Face Database (Bainbridge et al., 2013). Like conventional practice, we perform principal component analysis (PCA) on the faces the AAM was trained on, but unlike conventional practice, we do *not* reduce the number of principal components, resulting in a representation with $n = 10,764$ features.

**Model Evaluation.** Using leave-one-out cross-validation ($m = 52$), we evaluate the *prediction* MSE on held-out test data. More formally, for each held-out face $\mathbf{x}_i$, we predict a social decision ($\hat{y}_i$):

$$\hat{y}_i = \mathbf{x}_i^T \hat{\beta}, \tag{8}$$

where $\hat{\beta}$ is the minimum L2-norm estimator specified in Section 2. We then evaluate:

$$\text{mse}(\hat{\mathbf{y}}) = \frac{1}{m} \sum_{i=1}^{m} (y_i - \hat{y}_i)^2. \tag{9}$$

## 3.3 VALIDATION

Using the computational framework, as well as the responses collected in the social decision making study, we observe the prediction MSE on unseen test data is 1) within the standard error of the mean of the MMSE estimator, and 2) well below chance for a wide variety of social decision-making tasks, indicating the over-parameterized representation generalizes well across tasks in practice (Figure 3).

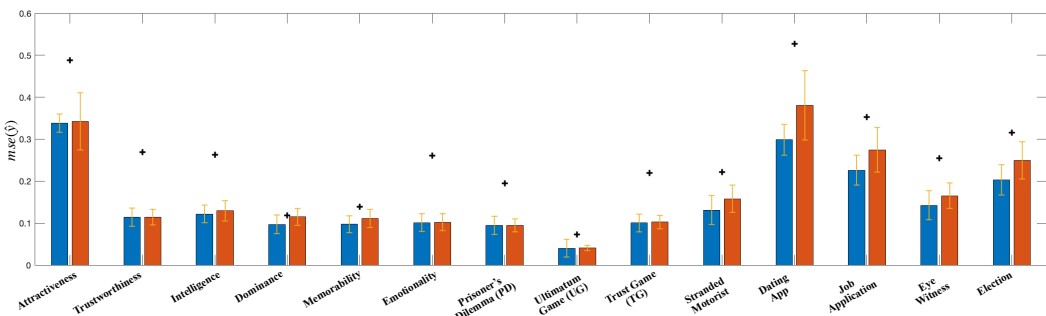

Figure 3: Experimental validation of our mathematical framework for several social decision tasks. For all tasks, the fully over-parameterized estimator (orange bars, right) is within the error bounds (standard error of the mean) of the MMSE estimator (blue bars, left) and below chance level (+; variance of collected responses) for all tasks (except dominance, which cannot be predicted better than chance by any model), indicating the over-parameterized representation generalizes well across tasks in practice. Note that the below-chance dominance estimator will not be used in any subsequent analysis.

## 4 APPLICATION: BEAUTY PENALTY AND TRUSTWORTHINESS HALO

Consistent with previous studies, we observe a strong positive correlation between collected attractiveness ratings and social decisions in both social scenarios and economic games (Figure 4A), indicating both an attractiveness halo and a beauty premium. We observe an even stronger positive correlation between trustworthiness ratings and social decisions (Figure 4A), which indicates both a trustworthiness halo (this typically refers to trait perception and decision making in social scenarios) and a trustworthy premium (this typically refers to decision making in economic games). However, the strong positive correlation between attractiveness and trustworthiness (CC = 0.53, p-value $\leq 0.001$) makes it impossible to separate the contributions of attractiveness and trustworthiness to the halo and premium effects from collected responses alone.

To tease these contributions apart, we use the mathematical and computational framework developed above. Using leave-one-out cross-validation, we compute predicted social decisions using orthogonalized estimators ($\hat{\beta}^{A \perp T}$ and $\hat{\beta}^{T \perp A}$), then compute the correlation between these predictions ($\hat{y}^{A \perp T}$ and $\hat{y}^{T \perp A}$) and social decisions. This orthogonalized estimators contain facial feature information unique to that trait (task) and not related to the other trait. To orthogonalize the estimators, we calculate the normalized projections of one estimator onto the other. We calculate the orthogonal projection of $\hat{\beta}_T$ onto $\hat{\beta}_A$ as

$$\hat{\beta}_{T \perp A} = \hat{\beta}_T - (\hat{\beta}_A \cdot \hat{\beta}_T)\hat{\beta}_A, \tag{10}$$

where $(\cdot)$ denotes the normalized dot product, and vice versa for the projection of $\hat{\beta}_A$ onto $\hat{\beta}_T$.

We observe (Figure 4A) attractiveness unrelated to trustworthiness is not significantly correlated with any social scenarios (except dating app), while trustworthiness unrelated to attractiveness is significantly correlated with all social scenarios (except dating app). This indicates the halo effect is driven by trustworthiness, rather than attractiveness, though it appears as an attractiveness effect due to the facial features that contribute to both attractiveness and trustworthiness.

We also observe (Figure 4B) attractiveness unrelated to trustworthiness is significantly *anti-correlated* with two out of three economic games, while trustworthiness unrelated to attractiveness is significantly correlated with all economic games. Once again, it seems what masquerades as an attractiveness effect is truly a trustworthiness effect, and that rather than inducing a beauty premium, attractiveness by itself (excluding those facial features also contributing to trustworthiness) induces a beauty *penalty*. Without teasing apart the two components using feature orthogonalization, the beauty penalty effect is masked by the strong beauty/trustworthiness premium effect.

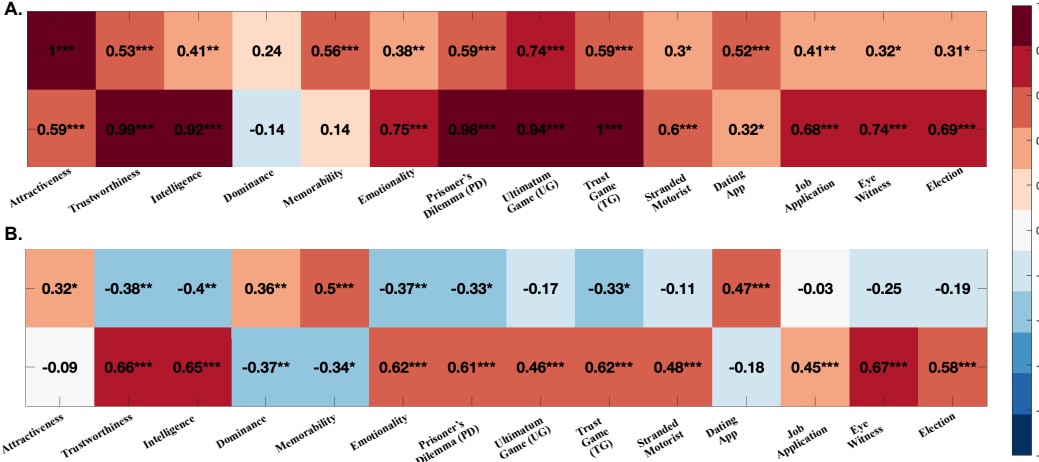

Figure 4: Heatmap of correlation coefficients (CCs) between social traits (cols. 1-6), social scenario decisions (cols. 10-14), economic games (cols. 7-9) with significance levels (*: p-value $\leq$ 0.05, **: p-value $\leq$ 0.01, ***: p-value $\leq$ 0.001). **(A)** Both attractiveness (first row) and trustworthiness (second row) are significantly positively correlated with all economic games and social scenarios (except trustworthiness and dating app). However, the strong positive correlation between attractiveness and trustworthiness (CC = 0.53, p-value < 0.001) makes it impossible to tease the contributions of these two traits apart using just the collected responses. **(B)** When attractiveness is unrelated to trustworthiness (first row), the significant positive correlation with social scenarios disappears (except dating app), dispelling an attractiveness halo effect. The positive correlation with two of the three economic games (PD and TG) also becomes significantly negative, indicating a beauty penalty. When trustworthiness is unrelated to attractiveness (second row), on the other hand, the significant positive correlation remains. This shows a trustworthiness halo effect in social scenario decisions, as well as a trustworthiness premium in economic games. Note that there is a significant anti-correlation between the attractiveness and unrelated trustworthiness (row 1, col. 2), indicating non-linear effects in the responses, which cannot be captured by the linear models.

Since AAM readily generates faces for any coordinates in the feature space, we can visualize the estimator (regression coefficient) axes and their orthogonalized versions (Figure 5). Visual inspection reveals both more attractive (top row) and trustworthy (bottom row) faces smile more, while less attractive faces also appear older. More interestingly, orthogonalizing the attractiveness estimator against the trustworthiness is no longer related to smiley-ness but appears anti-correlated with age (more youthful-looking faces are more attractive, which has previously been observed (Sutherland et al., 2013). Notably, the projections of the face stimuli used in the experiment along this dimension are indeed significantly correlated (CC = −0.29, p-value< 0.05), with previously collected age ratings of these faces Bainbridge et al. (2013), while these projections are significantly negatively correlated with ratings in economic games (Figure 4). To summarize, the above results imply that the youthfulness-related component of attractiveness drive a "beauty penalty" effect in economic games, while the facial features that drive both attractiveness and trustworthiness perception are what give rise to an attraction/trustworthiness halo. In addition, when we orthogonalize trustworthiness against attractiveness, a strong smiley-ness effect remains (just as in the unorthogoalized case), while the age effect mostly disappears. Moreover, we find that this residual component unrelated to attractiveness is still positively correlated with social scenario and economic games.

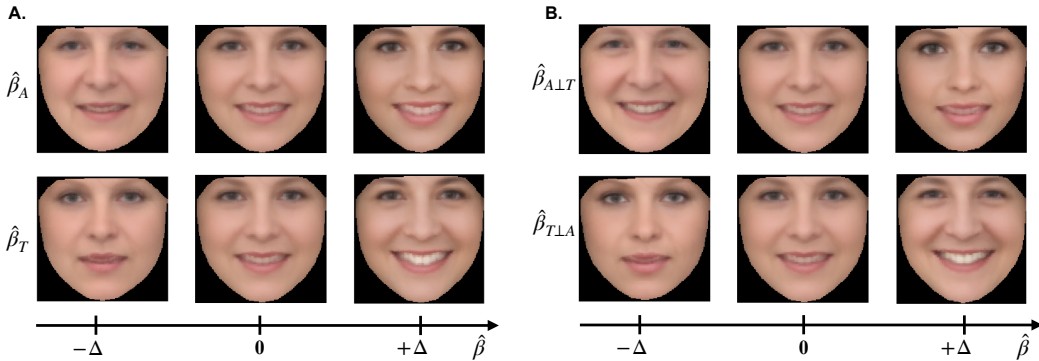

Figure 5: Face visualization along regression coefficient (estimators) directions without (A) and with (B) orthogonalization. Top row: attractiveness; bottom row: trustworthiness. The middle face in each triplet is the average face (corresponding to feature coordinates that average over all 52 faces used in the study). Visualization moves in equal steps along the estimator axes (left: negative direction, right: positive direction).

## 5 DISCUSSION

In this paper, we provided conditions under which an over-parameterized representation is guaranteed to yield optimal, as well as better than chance, generalization performance in a linear regression setting with hard constraints. We also provided exact expression for the estimator error, as well as an exact expression for how far the fully over-parameterized estimator is from the optimal hard-regularizer. We next validated the usefulness of our mathematical framework by applying it to a wide range of social decision-making tasks, in which the fully over-parameterized estimator performed within the error bounds (standard error of the mean) of the theoretically optimal estimator on all tasks. We then used this framework to show 1) the halo effect appears to arise from trustworthiness rather than attractiveness, and 2) trustworthiness unrelated to attractiveness induces a premium in economic games, while attractiveness unrelated to trustworthiness induces a penalty, indicating a trustworthiness premium and beauty penalty, which helps reconciling conflicting reports in the existing literature. Moreover AAM-based visualization indicated that the trustworthiness halo/premium is underpinned by smiley-ness and the beauty premium by youthfulness (through the component specifically important for attractiveness).

While some of the statistical analyses among traits, social scenarios, and economic games could have been done using only ratings, the ability of grounding those ratings in an image-computable and generative model representation is highly valuable. Without the latter, we wouldn't have been able to orthogonalize estimated regression coefficient vectors against one another (orthogonalization

makes no sense if feature vectors do not live in the same feature space), or visualize faces along those vectors. Such visualizations (Figure 5) reveal that both trustworthiness and trustworthiness unrelated to attractiveness appear highly correlated with smiling. This begs the question of how smiling, or emotional states such as happiness, contribute to the halo effect. Such data can be collected framework helps to identify concrete directions for future research endeavors. Having a universal, overparameterized representation that serve all tasks can assist with iterative scientific analysis and hypothesis generation, as new experiments are designed and data collected, and new conclusions are drawn.

One limitation of our framework is that it does not include contributions from non-linear components, which have been found to contribute to trait ratings, including attractiveness (Ryali & Yu, 2018; Todorov & Oosterhof, 2011) and trustworthiness (Todorov & Oosterhof, 2011). A further limitation of our study is that we only focused on female faces. There is evidence dominance and trustworthiness are rated using gender-based internal models (He & Yu, 2021), which could also be true of social decisions. In addition, a strong correlation between dominance and election success has been established in the literature for male faces (Berinsky et al., 2019). However, a preliminary analysis of dominance ratings collected in the social decision-making experiment reveals no such correlation for female faces (Figure 6), indicating different traits might contribute to halo effects for female and male faces. These questions remain exciting avenues for future work.

The more general limitations of our theoretical framework is that the optimality conditions only hold for hard regularizers, and that our error expressions are for estimator MSE rather than prediction MSE. Extending the general theoretical results to soft-regularizers (such as ridge- and lasso-regression) and the more practically useful prediction MSE are also exciting future directions.

### 5.1 RELATED THEORETICAL WORK IN OVER-PARAMETERIZED LINEAR REGRESSION

Our PCR approach might at first glance seem identical to that of Xu & Hsu (2019). However, while Xu & Hsu (2019) analyze what they call an "oracle" estimator, which uses the generative ("true") covariance matrix, we use the more classical version of PCR, which is based on the sample covariance matrix. This results in quite different behavior. For instance, there is no over-parameterized regime in PCR, as $m$ data points can be expressed by most $m$ linearly independent features ($m - 1$ when the data is centered). As such, there is no "second descent" in PCR. Xu & Hsu (2019) also noted that a full analysis that accounts for estimation errors in PCR remains open, though it is worth noting that an extensive analysis of the under-parameterized regime was done by Park (1981).

Also worth noting is that there seems to be a sharp divide between the"classical" under-parameterized and the "modern" over-parameterized regime in the literature, with an understanding of the latter "now only starting to emerge" (Belkin et al., 2020). We offer a different view by showing any over-parameterized representation has an "equivalent" under-parameterized representation, and as such, the over-parameterized regime can be fully understood in terms of the *under-parameterized* regime.

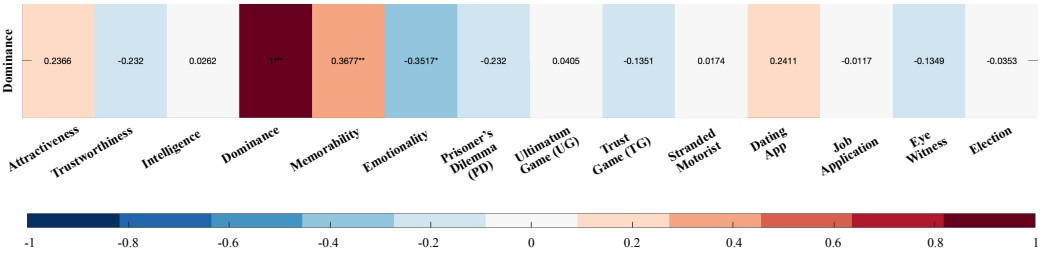

Figure 6: Heatmap of CCs between dominance and social traits (cols. 1-6), social scenario decisions (cols. 10-14), economic games (cols. 7-9) with significance levels (*: p-value $\leq 0.05$, **: p-value $\leq 0.01$, ***: p-value $\leq 0.001$). None of the CCs between dominance and social scenarios/economic games are significant, indicating dominance does not significantly contribute to decisions in these tasks. As there is an established correlation between dominance and election for male faces in the literature, this indicates different traits might contribute to halo effects for male and female faces.

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

## A  APPENDIX

### A.1  PROOFS

#### A.1.1  PRELIMINARIES

Recall, we consider an over-parameterized linear regression setting ($n \geq m$),

$$\mathbf{y} = \mathbf{X}\beta + \epsilon, \qquad \epsilon \sim N(0, \sigma_\epsilon^2) \tag{11}$$

with both $\beta \in \mathbb{R}^n$ and $\epsilon \in \mathbb{R}^m$ assumed to be zero-mean and i.i.d, and rank of $\mathbf{X} \in \mathbb{R}^{m \times n}$ denoted by $r$.

We use the fact that any matrix $\mathbf{X}$ can be written in terms of its singular value decomposition as $\mathbf{X} = \mathbf{U}\Sigma\mathbf{V}^T$, and $\mathbf{X}$ expressed in its principal component (PC) representation is simply $\mathbf{U}\Sigma$, which we denote by $\mathbf{Z}$.

We also use the PCR setting,

$$\mathbf{y} = \mathbf{Z}\theta + \epsilon, \qquad \epsilon \sim N(0, \sigma_\epsilon^2) \tag{12}$$

$$\hat{\theta} = \mathbf{Z}^\dagger \mathbf{y} \tag{13}$$

where $\mathbf{Z} \in \mathbb{R}^{m \times r}$ is the design matrix expressed in terms of its PCs and $\hat{\theta}$ is the PCR estimator.

**Note on terminology.** By the MMSE estimator, we mean the minimum mean squared error estimator among the class of hard regularizers subject to linear constraints (this does not include soft-regularizers, such as ridge and lasso estimators).

### A.1.2 PROOFS

**Lemma 1.** The PCR estimator with all PCs is an orthogonal transformation of the over-parameterized estimator with all features.

*Proof.* This follows from the definitions. Let $\hat{\theta}$ denote the PCR estimator and $\hat{\beta}$ denote the over-parameterized estimator. Then,

$$
\begin{aligned}
\hat{\beta} &= \mathbf{X}^\dagger \mathbf{y} \\
&= \mathbf{V}\mathbf{\Sigma}^\dagger \mathbf{U}^T \mathbf{y} \\
&= \mathbf{V}\mathbf{Z}^\dagger \mathbf{y} \\
&= \mathbf{V}\hat{\theta}.
\end{aligned}
$$

**Lemma 2.** Estimator MSE is invariant under orthogonal transformations.

*Proof.* Let $\beta$ denote the estimator and $\mathbf{V}$ denote an orthogonal matrix. Then,

$$
\begin{aligned}
\mathrm{mse}(\mathbf{V}\hat{\beta}) &= \mathrm{tr}\,\mathbb{E}[(\mathbf{V}\beta - \mathbf{V}\hat{\beta})(\mathbf{V}\beta - \mathbf{V}\hat{\beta})^T] \\
&= \mathrm{tr}\,\mathbb{E}[\mathbf{V}(\beta - \hat{\beta})(\beta - \hat{\beta})^T \mathbf{V}^T] \\
&= \mathrm{tr}\,\mathbf{V}\mathbb{E}[(\beta - \hat{\beta})(\beta - \hat{\beta})^T]\mathbf{V}^T \\
&= \mathrm{tr}\,\mathbb{E}[(\beta - \hat{\beta})(\beta - \hat{\beta})^T]\mathbf{V}^T\mathbf{V} \\
&= \mathrm{tr}\,\mathbb{E}[(\beta - \hat{\beta})(\beta - \hat{\beta})^T] \\
&= \mathrm{mse}(\hat{\beta})
\end{aligned}
$$

**Lemma 3.** The MSE of the PCR estimator with all PCs equals the MSE of the over-parameterized estimator with all features.

*Proof.* This follows from Lemmas 1 and 2. Let $\hat{\theta}$ denote the PCR estimator and $\hat{\beta}$ denote the over-parameterized estimator. Then,

$$
\mathrm{mse}(\hat{\theta}) \stackrel{(1)}{=} \mathrm{mse}(\mathbf{V}\hat{\theta}) \stackrel{(2)}{=} \mathrm{mse}(\hat{\beta}).
$$

**Lemma 4.** The MSE of the minimum MSE PCR estimator lower bounds the MSE of the over-parameterized estimator with all $n$ features.

*Proof.* This follows from the definition of the minimum MSE, as well as Lemma 3. Denote the minimum MSE PCR estimator as $\hat{\theta}^*$, the PCR estimator with all PCs as $\hat{\theta}$ and the over-parameterized estimator with all features as $\hat{\beta}$. Then,

$$
\mathrm{mse}(\hat{\theta}^*) \stackrel{(\mathrm{def})}{\leq} \mathrm{mse}(\hat{\theta}) \stackrel{(3)}{=} \mathrm{mse}(\hat{\beta}).
$$

**Lemma 5.** The MSE of the minimum MSE PCR estimator lower bounds the MSE of an over-parameterized, feature reduced estimator.

*Proof.* This follows from Lemma 3, as well as Park (1981). Let $\mathbf{X}_n$ be the original design matrix with all $n$ features, and denote the minimum PCR estimator of this design matrix as $\hat{\theta}^*$. Let $\mathbf{X}_p$ be a feature reduced design matrix, expressed in any of the $p < n$ original features, and denote the $p$ dimensional estimator of $\mathbf{X}_p$ as $\hat{\alpha}$. Denote $\hat{\phi}^*$ and $\hat{\phi}$ as the minimum MSE PCR estimator and PCR estimator with all PCs respectively. Note that these estimators are under-parameterized estimators. It then follows that,

$$\text{mse}(\hat{\alpha}) \overset{(3)}{=} \text{mse}(\hat{\phi}) \overset{\text{(def)}}{\geq} \text{mse}(\hat{\phi}^*) \overset{\text{Park (1981)}}{\geq} \text{mse}(\hat{\theta}^*).$$

**Theorem 1.** The MSE of the minimum MSE PCR estimator lower bounds the MSE in both the over-parameterized and under-parameterized regimes.

*Proof.* This follows as the minimum MSE PCR estimator lower bounds the MSE of under-parameterized estimators (Park, 1981), as well as over-parameterized estimators (lemmas 4 and 5). $\qquad \square$

**Park 1**. If the $p$-th singular value ($s_p$) of the design matrix $\mathbf{X}$ satisfies,

$$s_p^2 \geq \frac{\sigma_\epsilon^2}{||\beta||^2/r}, \tag{14}$$

then $\hat{\theta}_p$ is the MMSE PCR estimator (Park, 1981).

*Proof.* See Park (1981). $\qquad \square$

**Estimator Condition 2.** If the smallest singular value ($s_r$) of the design matrix $\mathbf{X} \in \mathbb{R}^{m \times n}$ satisfies,

$$s_r^2 \geq \frac{\sigma_\epsilon^2}{||\beta||^2/r}, \tag{15}$$

then the over-parameterized estimator with all $n$ features is the minimum MSE estimator.

*Proof.* This follows from Park 1, as well as Theorem 1. Denote the minimum MSE PCR estimator as $\hat{\theta}^*$, the PCR estimator with all PCs as $\hat{\theta}$, and the over-parameterized estimator with all $n$ features as $\hat{\beta}$, and suppose the threshold is satisfied for the $r$-th singular value. Then,

$$\text{mse}(\hat{\theta}^*) = \text{mse}(\hat{\theta}) = \text{mse}(\hat{\beta}), \tag{16}$$

which we know from Theorem 1 lower bounds the MSE in both the under-parameterized and over-parameterized regimes. As such, $\hat{\beta}$ is an MMSE estimator. $\qquad \square$

**Error Expression 1**. The MSE of the over-parameterized estimator without feature reduction is given by

$$\text{mse}(\hat{\beta}) = \sigma_\epsilon^2 \sum_{i=1}^{r} \frac{1}{s_i^2}, \tag{17}$$

where $\sigma_\epsilon^2$ is the noise variance, and $s_1, .., s_r$ the singular values of the design matrix $\mathbf{X}$.

*Proof.* Note that $\beta$ can be written in terms of $\mathbf{X}$, $\mathbf{y}$, and $\epsilon$, as $\beta = \mathbf{X}^\dagger(\mathbf{y} - \epsilon)$.

Then,

$$
\begin{aligned}
mse(\hat{\beta}) :&= \text{tr}\,\mathbb{E}[(\beta - \hat{\beta})(\beta - \hat{\beta})^T] \\
&= \text{tr}\,\mathbb{E}[(\mathbf{X}^\dagger \epsilon)(\mathbf{X}^\dagger \epsilon)^T] \\
&= \text{tr}\,\mathbb{E}[\mathbf{X}^\dagger \epsilon \epsilon^T \mathbf{X}^{\dagger T}] \\
&= \text{tr}(\mathbf{X}^\dagger \mathbb{E}[\epsilon \epsilon^T]\mathbf{X}^{\dagger T}) \\
&= \sigma_\epsilon^2 \,\text{tr}(\mathbf{X}^\dagger \mathbf{X}^{\dagger T}) \\
&= \sigma_\epsilon^2 \,\text{tr}(\mathbf{U}\mathbf{\Sigma}^{\dagger^2}\mathbf{U}^T) \\
&= \sigma_\epsilon^2 \,\text{tr}(\mathbf{\Sigma}^{\dagger^2}\mathbf{U}^T\mathbf{U}) \\
&= \sigma_\epsilon^2 \,\text{tr}\,\mathbf{\Sigma}^{\dagger^2} \\
&= \sigma_\epsilon^2 \sum_{i=1}^{r} \frac{1}{s_i^2}.
\end{aligned}
$$
$\qquad \square$

**Lemma 6.** The variance of the true parameter vector $\hat{\beta}$ is given by

$$\sigma_\beta^2 = (\sigma_y^2 - \sigma_\epsilon^2) \sum_{i=1}^{r} \frac{1}{s_i^2}, \tag{18}$$

where $\sigma_y^2$ is signal variance, $\sigma_\epsilon^2$ the noise variance, and $s_1, .., s_r$ are the singular values of the design matrix.

*Proof.*

$$
\begin{aligned}
\sigma_\beta^2 :&= \operatorname{tr} \mathbb{E}[\beta\beta^T] \\
&= \operatorname{tr} \mathbb{E}[\mathbf{X}^\dagger(\mathbf{y}-\epsilon)(\mathbf{y}-\epsilon)^T\mathbf{X}^{\dagger T}] \\
&= \operatorname{tr}(\mathbf{X}^\dagger\mathbb{E}[(\mathbf{y}-\epsilon)(\mathbf{y}-\epsilon)^T]\mathbf{X}^{\dagger T}) \\
&= \operatorname{tr}\mathbf{X}^\dagger(\mathbb{E}[\mathbf{y}\mathbf{y}^T]-\mathbb{E}[\epsilon\epsilon^T])\mathbf{X}^{\dagger T} \\
&= (\sigma_y^2 - \sigma_\epsilon^2)\operatorname{tr}(\mathbf{X}^\dagger\mathbf{X}^{\dagger T}) \\
&= (\sigma_y^2 - \sigma_\epsilon^2)\sum_{i=1}^r \frac{1}{s_i^2}.
\end{aligned}
$$

**Estimator Condition 1.** If the noise-variance in $\mathbf{y}$ ($\sigma_\epsilon^2$) is less than or equal to half the signal-variance in $\mathbf{y}$ ($\sigma_y^2$), then $\hat{\beta}$ is an above chance estimator, i.e.

$$
\sigma_\epsilon^2 \le \frac{\sigma_y^2}{2} \quad \Longleftrightarrow \quad \operatorname{mse}(\hat{\beta}) \le \sigma_\beta^2, \tag{19}
$$

where $\sigma_\beta^2$ is the variance of the parameter vector $\beta$.

*Proof.* This follows from Error Expression 1 and Lemma 6.

$$
\begin{aligned}
\operatorname{mse}(\hat{\beta}) \quad &\le \sigma_\beta^2 \\
\Longleftrightarrow \quad \sigma_\epsilon^2\sum_{i=1}^r\frac{1}{s_i^2} &\le (\sigma_y^2-\sigma_\epsilon^2)\sum_{i=1}^r\frac{1}{s_i^2} \\
\Longleftrightarrow \quad \sigma_\epsilon^2 \quad &\le \sigma_y^2 - \sigma_\epsilon^2 \\
\Longleftrightarrow \quad 2\sigma_\epsilon^2 \quad &\le \sigma_y^2 \\
\Longleftrightarrow \quad \sigma_\epsilon^2 \quad &\le \frac{\sigma_y^2}{2}
\end{aligned}
$$

**Lemma 7.** The MSE of the feature reduced PCR estimator $\hat{\theta}_p$ (with $p \le r$ coefficients) is given by,

$$
\operatorname{mse}(\hat{\theta}_p) = \frac{||\beta||^2}{r}(r-p) + \sigma_\epsilon^2\sum_{i=1}^p\frac{1}{s_i^2} \tag{20}
$$

*Proof.* Recall that $\theta$ can be written in terms of $\mathbf{Z}$, $\mathbf{y}$, and $\epsilon$, as $\theta = \mathbf{Z}^\dagger(\mathbf{y}-\epsilon)$. First note that the feature reduced PCR estimator $\hat{\theta}_p$ is given by,

$$
\begin{aligned}
\hat{\theta}_p :&= \mathbf{Z}_p^\dagger\mathbf{y} \\
&= \mathbf{Z}_p^\dagger(\mathbf{Z}\theta + \epsilon) \\
&= \mathbf{Z}_p^\dagger\mathbf{Z}\theta + \mathbf{Z}_p^\dagger\epsilon \\
&= \mathbf{\Sigma}_p^\dagger\mathbf{U}^T\mathbf{U}\mathbf{\Sigma}\theta + \mathbf{Z}_p^\dagger\epsilon \\
&= \mathbf{\Sigma}_p^\dagger\mathbf{\Sigma}\theta + \mathbf{Z}_p^\dagger\epsilon \\
&= \mathbf{I}_p\theta + \mathbf{Z}_p^\dagger\epsilon,
\end{aligned}
$$

where $\mathbf{I}_p$ is an $m \times r$ dimensional matrix with ones on the diagonal for the first $p$ entries and zeros on the remainder.

It then follows that,

$$
\begin{aligned}
\mathrm{mse}(\hat{\theta}_p) : &= \mathrm{tr}\,\mathbb{E}[(\theta - \hat{\theta}_p)(\theta - \hat{\theta}_p)^T] \\
&= \mathrm{tr}\,\mathbb{E}[((\mathbf{I}_r - \mathbf{I}_p)\theta + \mathbf{Z}_p^\dagger \epsilon)(\epsilon^T \mathbf{Z}_p^{\dagger T} + \theta^T(\mathbf{I}_r - \mathbf{I}_p)^T)] \\
&= \mathrm{tr}\,\mathbb{E}[(\mathbf{I}_r - \mathbf{I}_p)\theta\theta^T(\mathbf{I}_r - \mathbf{I}_p)^T] + \mathrm{tr}\,\mathbb{E}[\mathbf{Z}_p^\dagger \epsilon\epsilon^T \mathbf{Z}_p^{\dagger T}] \\
&= \mathrm{tr}(\mathbf{I}_r - \mathbf{I}_p)\mathbb{E}[\theta\theta^T](\mathbf{I}_r - \mathbf{I}_p)^T + \sigma_\epsilon^2 \,\mathrm{tr}\,\boldsymbol{\Sigma}_\mathbf{p}^\dagger \\
&= \frac{||\theta||^2}{r}\,\mathrm{tr}(\mathbf{I}_r - \mathbf{I}_p)^2 + \sigma_\epsilon^2 \,\mathrm{tr}\,\boldsymbol{\Sigma}_\mathbf{p} \\
&= \frac{||\theta||^2}{r}(r - p) + \sigma_\epsilon^2 \sum_{i=1}^{p} \frac{1}{s_i^2} \\
&= \frac{||\beta||^2}{r}(r - p) + \sigma_\epsilon^2 \sum_{i=1}^{p} \frac{1}{s_i^2}.
\end{aligned}
$$

**Error Expression 2.** Suppose the MMSE PCR estimator $\hat{\theta}^*$ has $p \leq r$ components. Then the difference in MSEs between the MMSE estimator and the fully over-parameterized estimator is given by,

$$
mse(\hat{\beta}) - mse(\hat{\theta}^*) = \sigma_\epsilon^2 \sum_{i=p+1}^{r} \frac{1}{s_i^2} - \frac{||\beta||^2}{r}(r - p). \tag{21}
$$

*Proof.* This follows from Lemma 7 as well as Error Expression 1.

$$
\begin{aligned}
\mathrm{mse}(\hat{\beta}) - \mathrm{mse}(\hat{\theta}^*) &= \sigma_\epsilon^2 \sum_{i=1}^{r} \frac{1}{s_i^2} - \left(\frac{||\beta||^2}{r}(r - p) + \sigma_\epsilon^2 \sum_{i=1}^{p} \frac{1}{s_i^2}\right) \\
&= \sigma_\epsilon^2 \left(\sum_{i=1}^{r} \frac{1}{s_i^2} - \sum_{i=1}^{p} \frac{1}{s_i^2}\right) - \frac{||\beta||^2}{r}(r - p) \\
&= \sigma_\epsilon^2 \sum_{i=p+1}^{r} \frac{1}{s_i^2} - \frac{||\beta||^2}{r}(r - p)
\end{aligned}
$$

