# OpenReview forum: "Leveraging Double Descent for Scientific Data Analysis: Face-Based Social Behavior as a Case Study"
_ICLR.cc/2023/Conference — Submitted to ICLR 2023_

### Official Review · Reviewer_ZkzX · 2022-10-24

**Confidence:** 5
**Clarity, Quality, Novelty And Reproducibility:** See Strength And Weaknesses.
**Correctness:** 3
**Technical Novelty And Significance:** 3
**Empirical Novelty And Significance:** 3
**Recommendation:** 3

**Strength And Weaknesses:**

The organization is clear and easy to follow.

My concerns regarding this paper are as below.
1. Limited novelty. The main contributions of this paper are not enough for ICLR.
2. Some related works are missing, e.g., Integrated Face Analytics Networks through Cross-Dataset Hybrid Training.

**Summary Of The Paper:**

This paper leverages double descent for scientific data analysis, with face-based social behavior as a case study.

**Summary Of The Review:**

See Strength And Weaknesses.

---

### Official Review · Reviewer_nFuG · 2022-10-25

**Confidence:** 5
**Correctness:** 2
**Technical Novelty And Significance:** 1
**Empirical Novelty And Significance:** 1
**Recommendation:** 1

**Clarity, Quality, Novelty And Reproducibility:**

This paper has lots of mathematical formula and proofs, but essentially it's about linear regression. I don't know if that's necessary given the focus of this paper. As mentioned above, I'm skeptical about the novelty and relevance of this paper. Another key question is about the choice of AAM. I'm not against using old methods, but there's no rationale provided regarding this choice despite numerous results and evidence showing the advantage of deep neural networks on this very topic published in the last decade. I'm also not sure about the design of the entire study, why not just use manual ratings? Why do we use a computational model when we have only 72 images? Also, I don't know how come AAM features are of length over 10k. They are usually much shorter.

**Strength And Weaknesses:**

I don't understand the main contributions of this paper. This paper doesn't feel like a paper about AI. It would be much more appropriate in psychology venues. But even then, I don't think this paper has original contributions. Both in psychology and computer vision literature, the use of CV/ML models has been explored extensively. These methods include traditional methods such as HOG and SVM as well as deep neural networks. Any of these papers are discussed in the manuscript. If this paper is about demonstrating that one can use automated methods for studies in face perception, many papers already exist. This paper is not about proposing a new method. Overall it's unclear what this paper tries to achieve or how it relates to the ICLR community.

**Summary Of The Paper:**

This paper studies correlations between perceived facial traits and social preference. This is a well studied topic in social psychology. The authors collected manual annotations on various facial traits for 72 images from undergraduate students (by the way the dataset used in the paper already provides such ratings for thousands of images). The paper uses AAM to extract features from facial images and fits a linear regression to estimate the coefficients. Some findings about relationship between social traits are reported.

**Summary Of The Review:**

I don't think this paper is relevant to AI or ICLR community. The quality and novelty are very limited too.

---

### Official Review · Reviewer_9oX1 · 2022-10-25

**Confidence:** 3
**Correctness:** 4
**Technical Novelty And Significance:** 2
**Empirical Novelty And Significance:** 2
**Recommendation:** 3

**Clarity, Quality, Novelty And Reproducibility:**

The methodology part is potentially novel in the estimator conditions presented, which however may not be applicable to real world data

the application part uses the idea of AAM's - when not reducing the dimensionality after PCA - in order to use as design matrix in this specific linear regression setting.  Insights from the experiment may be novel to the relevant community

**Strength And Weaknesses:**

The paper presents interesting insights in the experimental part based on the study, relating various traits to social decision making tasks.  The proposed framework is essentially a linear regression setting, however as the authors also mention - some degree of non-linearity may be necessary for modelling these mappings due to the nature of the data.  At the same time, although some theoretical conditions are described, the authors mention that the experiment is used to verify the framework since real-world data does not adhere to most of these conditions, however the only support that the conditions are satisfied is  the MSE Plot - Fig 3.   The conclusions on beauty penalty and trustworthiness are quite interesting.  The concept of orthogonalized estimators is also interesting in this context and appears to offer insight into the relationship between them.

**Summary Of The Paper:**

The authors propose a linear framework that relies on overparametrized representations in order to facilitate analysis of features, and their impact to multiple response variables.  The authors motivate this in underdetermined problems, with a larger number of features than data points.  The authors show some conditions for a linear regression framework with an overparameterized full rank design matrix, e.g., when the model constitutes an above chance estimator.  They then proceed to describe an experiment, which is used to validate the hypothesis - given that such conditions are hard to be satisfied by real-world data.  The experiment is based on data from a study including ~ 600 participants on social decision making tasks, with outliers excluded automiatically.  72 images were used in the study.  Computationally, the authors train an AAM, however they used the full set of principal components to get an overparameterized representation.  They then proceed to analyse the results, and find correlations between traits and the social decision making tasks.  Some related insights are presented.

**Summary Of The Review:**

I think that some of the insights presented on social decision making can be interesting and feature representations are central to this paper.   However *in my opinion which may be wrong* technically the paper seems to be oriented more at the intersection of statistics and psychology.   There are also no comparisons to any other methods that may facilitate some interpretability - e.g., what if a different representation than full PCA on AAMs was used?

---

### Official Review · Reviewer_pCsM · 2022-10-28

**Confidence:** 2
**Clarity, Quality, Novelty And Reproducibility:** All well written and perfectly clear.
**Correctness:** 4
**Technical Novelty And Significance:** 3
**Empirical Novelty And Significance:** 3
**Recommendation:** 5

**Details Of Ethics Concerns:**

Involvement of humans and beauty are a potential critical  topic.

**Strength And Weaknesses:**

This is an interesting study from a social perspective. The findings bear merit and all is well described. The mathematical model is of interest.

It is, to the reviewer, however, largely unclear what the core relation to ICLR is.

**Summary Of The Paper:**

The authors analyse via mathematical modelling and 600+ participating students the effects of facial beauty. They find both rewards (including based on perceived trustworthiness) and penalties from +/- beauty.

**Summary Of The Review:**

A great paper for social sciences, psychology, or other venues - the reviewer does not see sufficient connection to ICLR, though.

---

### Decision · Program_Chairs · 2023-01-20

**Decision:**

Reject

**Justification For Why Not Higher Score:**

The main concern is that the paper is out of the scope of ICLR.

**Justification For Why Not Lower Score:**

N/A

**Metareview: Summary, Strengths And Weaknesses:**

This paper uses the concept of double descent to develop a method to analyze features and their impact to variables. The method is used to study human ratings of social traits on a new face dataset. Two of the reviewers pointed out the interest of the insights revealed by the experiments, which can be relevant for the scientific community on psychology. However, the reviewers agree the paper is out of the scope of ICLR, since the main contribution are the findings but not the technique. The authors did not submit any feedback.